# Measurement of the Acoustic Relaxation Absorption Spectrum of CO_2_ Using a Distributed Bragg Reflector Fiber Laser

**DOI:** 10.3390/s23104740

**Published:** 2023-05-14

**Authors:** Kun Shen, Jixian Yuan, Min Li, Xiaoyan Wen, Haifei Lu

**Affiliations:** 1School of Science, Wuhan University of Technology, Wuhan 430070, China; 2Liangyuan Institute of Science and Technology Information, Shangqiu 476000, China

**Keywords:** acoustic relaxation absorption, gas relaxation acoustics, ultrasonic sensor, DBR fiber laser, carbon dioxide detection

## Abstract

Reconstruction of the acoustic relaxation absorption curve is a powerful approach to ultrasonic gas sensing, but it requires knowledge of a series of ultrasonic absorptions at various frequencies around the effective relaxation frequency. An ultrasonic transducer is the most widely deployed sensor for ultrasonic wave propagation measurement and works only at a fixed frequency or in a specific environment like water, so a large number of ultrasonic transducers operating at various frequencies are required to recover an acoustic absorption curve with a relative large bandwidth, which cannot suit large-scale practical applications. This paper proposes a wideband ultrasonic sensor using a distributed Bragg reflector (DBR) fiber laser for gas concentration detection through acoustic relaxation absorption curve reconstruction. With a relative wide and flat frequency response, the DBR fiber laser sensor measures and restores a full acoustic relaxation absorption spectrum of CO_2_ using a decompression gas chamber between 0.1 and 1 atm to accommodate the main molecular relaxation processes, and interrogates with a non-equilibrium Mach-Zehnder interferometer (NE-MZI) to gain a sound pressure sensitivity of −45.4 dB. The measurement error of the acoustic relaxation absorption spectrum is less than 1.32%.

## 1. Introduction

Ultrasonic gas sensing technology is an integration technique of gas sensing with an ultrasonic sensor that has two main technical branches depending either on acoustic velocity or acoustic relaxation absorption.

As early as the late 19th century, Fritz Haber used sound velocity change to assess the presence of hydrogen in the air during underground mining or a burning gas whistle [1,2]. Polturak used the acoustic travel time (time of flight) method to measure sound in a propane and ethylene binary gas mixture and reached a relative high sensitivity of 0.008% [3]. The Cassini-Huygens spacecraft used sound velocity to calculate changes in methane concentration in Titan’s atmosphere as a function of altitude [4]. These methods are only applicable to the concentration detection of binary gas mixtures due to considering a single sound propagation characteristic parameter, velocity, and become complicated when more than three components are involved in mixed gases. Since sound velocity cannot provide enough information, another important parameter, the sound absorption coefficient, came into sight. Tours et al. [5] proposed a tri-component gas detection approach using both sound velocity and acoustic absorption in 1985, but considered only classical absorption, which concerns the macroscopic energy consumption of gas in the environment, rather than acoustic relaxation absorption, which concerns the energy transformation between the external energy (translational freedom energy) and internal energy (rotational freedom kinetic energy and vibration freedom energy) of a molecule led by molecular relaxation processes (i.e., the energy of gas transfers from the external energy to the internal energy and has a characteristic frequency for a specific gas). In 2003, Phillips et al. [6] introduced the theory of simultaneous measurement of sound velocity and attenuation to achieve the measurement of ternary gas concentration. The product of sound velocity and sound absorption coefficients is called the acoustic relaxation absorption coefficient, which can be used for three- or four-element gas detection.

Measurement of the acoustic transmission spectrum is the other technical solution for ultrasonic gas sensing, which collects relevant information about multi-component gases from both the acoustic absorption and acoustic velocity spectra; i.e., a practical method for the measurement of the acoustic absorption spectrum is a prerequisite for gas sensing. The absorption curves of most excitable gases (non-mono-atomic or molecular gases) cover the frequency range of 10~10^8^ Hz at room temperature and pressure [7]. Current consumer ultrasonic transducers are only launched at one frequency, aside from a few with kilohertz bandwidth. A large number of acoustic transducers at different frequencies are required to cover the entire frequency spectral line to be measured [8]. Therefore, a wide-band ultrasonic sensor with a flat response would be preferred. Since the ultrasonic transmitter is not able to be replaced at present, an alternative approach to adjusting the ambient pressure, which is inversely proportional to the relaxation frequency of gas molecules [9], to obtain the whole sound propagation spectrum with a transmitter and a DBR fiber laser is proposed in this paper. 

Compared with piezoelectric transducers, cantilever beams, tuning forks, and other ultrasonic sensors, the newly emerged optical fiber ultrasonic sensors have the advantages of small size, wide bandwidth, and high sensitivity [10]. In 2017, Gang et al. [11] proposed a miniature compact Michelson interferometer, which completed a highly sensitive detection of 100~300 kHz ultrasound waves using a gold film deposit fiber tip. Another Fabry-Perot interferometer (FPI) ultrasonic transducer achieved noise equivalent sound pressure as low as 2 Pa and a bandwidth of −6 dB greater than 22.5 MHz, but consistency is an issue for FPI ultrasonic sensors. Moreover, its demodulation usually depends on a narrow linewidth tunable laser with a high cost [12]. An ultrasonic sensor based on a microfiber MZI is proposed [13], whose noise equivalent sound pressure and the highest detectable frequency reached 0.15 kPa and 40 MHz, respectively. Restoration of the acoustic absorption curve of gas requires measuring the ultrasonic waves in the kHz~MHz frequency range, which increasingly attenuate in gas with frequency and require a more sensitive sensor to respond to the weak ultrasonic signal.

Compared with other optical ultrasonic sensors, a DBR fiber laser sensor has the merits of a narrow line width (usually less than a few kilohertz), high output power, and stability, which would derive further benefits from interferometric demodulation, low noise, and high sensitivity. A DBR fiber laser composed of a piece of active optical fiber between two fiber gratings written in pairs is proposed as the ultrasonic sensing element, which is capable of detecting the wavelength shift affected by weak sound pressure. The applied ultrasonic wave can be recognized by continuing to detect the optical frequency variation, which is capable of a noise equivalent sound pressure as low as 45 Pa and a frequency up to 100 MHz using phase demodulation technology based on a non-equilibrium M-Z interferometer (NE-MZI) [14]. Guan Baiou et al. reported a photoacoustic imaging application of fiber laser bent into different diameters to achieve ultrasonic response with different spatial resolution [15]. In 2022, Guan’s team developed a miniaturized photoacoustic endoscope system with a noise equivalent pressure density (NEPD) below 1.5 mPa/Hz^1/2^ and a measurement range from 5 to 25 MHz [16]. As a highly sensitive candidate, the DBR fiber laser is also promising for marine applications such as pressure, tsunami, and earth dynamics [17].

This paper presents a DBR fiber laser sensor of gas concentration by restoring the gas acoustic propagation curve, which makes up for the shortcomings of the fixed frequency and relative low sensitivity of current consumer ultrasonic transducers. In Section 2, the ultrasonic response and sensing model of the DBR fiber laser are discussed before applying to the relaxation absorption coefficient measurement and calibration. Section 3 experimentally demonstrates the reconstruction process of the relaxation absorption curve and concentration measurement of CO_2_ using the DBR fiber laser ultrasonic sensor. 

## 2. Ultrasonic Sensing Model of the DBR Fiber Laser

### 2.1. Response of the DBR Fiber Laser to Ultrasonic Wave

A DBR fiber laser consists of a pair of fiber Bragg gratings (FBGs) and a piece of Er-doped fiber in between that works as a sensing unit. The process of applying a soundwave to the DBR fiber laser changes the cavity length and, in turn, the laser wavelength, which involves two physical effects: elastic deformation and the photoelastic effect of the DBR fiber laser. The induced strain of the DBR fiber laser under an ultrasonic wave needs to be figured out before sensor modeling. In this work, the displacement of the DBR fiber laser is obtained by solving the partial differential equation of the one-dimensional vibration of the optical fiber and numerically proved by finite element simulation.

Figure 1 illustrates the gas detection scheme proposed in this paper. Transmitting transducers generate ultrasonic waves of several specific frequencies, which are received by the DBR fiber laser.

As a kind of mechanical wave, the ultrasonic wave acting on the DBR fiber laser follows the wave theory of a string and generates longitudinal stress waves propagating along the fiber axis, which produce axial strain on the DBR fiber laser. Considering the DBR fiber laser as a thin rod consisting of several segments, longitudinal displacement of each segment leads to compression or elongation of its adjacent segments.

Assuming the ultrasonic wave is a plane wave emitted from the transducer, the acoustic sound pressure in the *z* direction (perpendicular to the fiber axis) *P*(*z*) can be written as follows:(1)$$P\left(z\right)=P_{0}e^{-\alpha z}$$
where *P*_0_ is the sound pressure at the transmitter and α is the attenuation coefficient.

Assuming the DBR fiber laser is an uniform bar, and *u*(*x*, *t*) is the displacement perpendicular to the fiber axis at position *x* and time *t* of the DBR fiber laser. The vibration equation has an expression as follows:(2)$$\frac{\partial^{2}u}{\partial t^{2}}=v^{2}\frac{\partial^{2}u}{\partial x^{2}}+P\left(z\right)$$

In Equation (2), *v* is the velocity of sound in the optical fiber. The displacement on the DBR fiber laser caused by the continuous force of ultrasonic waves presents as *u*(*x, t*), and the sound pressure *P* modulates the sensor’s geometry, which leads to a wavelength shift of the DBR fiber laser.
(3)$$\begin{array}{l} u\left(x,t\right)=\frac{4v_{0}L}{\pi^{2}v}\sum_{m=1}^{\infty }\frac{1}{m^{2}}\sin\frac{m\pi x_{0}}{L}\sin\frac{m\pi d}{L}\sin\frac{m\pi vt}{L}\sin\frac{m\pi x}{L}\\ +\sum_{m=1}^{\infty }\left[\frac{1}{m\pi v}\int_{0}^{t}P_{m}\left(\tau \right)\sin\frac{m\pi v\left(t-\tau \right)}{L}d\tau \right]\sin\frac{m\pi x}{L} \end{array}$$

Equation (3) is the one-dimensional vibration equation of the DBR fiber laser, in which *m* is a positive integer, *d* is the fiber segment length under an ultrasonic wave, *L* is the length of the DBR fiber laser, *v*_0_ is the initial velocity of the fiber vibration, and *P_m_*(τ) is the sound pressure acting on the DBR fiber laser, which is proportional to *P*(*z*) in Equation (1).

Figure 2 illustrates the displacement distribution of the DBR fiber laser under ultrasonic sound pressure of 2 Pa and 25 kHz according to Equation (3).

A 25 kHz ultrasonic wave is launched perpendicularly to the axis of the DBR fiber laser, the two ends of which are fixed. The ultrasonic wave acting on the DBR fiber laser is set at 2 Pa and covers a fiber segment length of 20 mm. 

Figure 2a plots the displacement distribution of the DBR fiber laser by solving the vibration equation as well as the finite element simulation results (Figure 2b). The center node has a maximum displacement of about 1.5 × 10^−17^ m for the DBR fiber laser length of 48 mm. 

The wavelength shift of the DBR fiber laser caused by the ultrasonic wave can be deduced from the obtained displacement. Since *P* is a function of the ultrasonic frequency, the wavelength shifts of the DBR fiber laser under 25 kHz (black square), 40 kHz (red dot), 112 kHz (blue up triangle), 200 kHz (green down triangle), and 300 kHz (purple diamond) ultrasonic waves are compared in Figure 3. In Figure 3, the maximum wavelength shift of the DBR fiber laser increases with the ultrasonic source strength, which is proportional to the sound pressure applied to the DBR fiber laser at a specific frequency, and decreases while the ultrasonic frequency increases. 

The DBR fiber laser is essentially the sensing unit and is subjected to sound pressure. The cavity length changes with the applied ultrasonic wave, which results in the wavelength shift of the DBR fiber laser. In Figure 3, the intensity of the ultrasonic wave is indicated by the source strength. For ultrasonic waves of a certain frequency, the wavelength shift of the DBR fiber laser is proportional to the intensity of the sound source and decreases as the frequency of the sound source increases, which results from the faster attenuation of the ultrasonic wave at higher frequencies. Since the better linearity of the DBR fiber laser sensing system brings with it a smaller systematic error in gas detection, the response linearity of the DBR fiber laser sensing system is experimentally demonstrated in Section 2.2.

The DBR fiber laser has two specific frequencies to be determined: the resonance frequency and the theoretical maximum response frequency, which are limited by the geometry of the DBR fiber laser. Using the equation of string vibration, resonant frequency $f_{m}=mv/2L,m=1,2,3\ldots$ When the density and Young’s modulus of the communication fiber are chosen, the sound velocity in the DBR fiber laser is about 5570 m/s. Thus, the frequency response of the fundamental mode is around 58 kHz. The fundamental vibration mode has the maximum amplitude, which is inversely proportional to m square (*m* is the order of the harmonic mode). On the other hand, only the axial strain needs to be considered when the ultrasonic wavelength is much larger than the diameter of the DBR fiber laser (125 μm). Therefore, the frequency response of the DBR fiber laser in CO_2_ reaches above 2 MHz (the wavelength of the sound wave and the response bandwidth increase while the molar mass of the gas decreases), which fully meets the experimental requirements.

The output light frequency *f_light_* of the DBR fiber laser depends on its cavity geometry, written as follows:(4)$$f_{light}=Q\frac{c}{2nL}$$
where *Q* is the positive integer, *c* is the velocity of light, *n* is the refractive index of the fiber, and *L* is the length of the DBR fiber laser. 

The relation between output frequency, cavity length, and refractive index of the DBR fiber laser has an expression as the derivative of Equation (4):(5)$$\frac{\Delta f_{light}}{f_{light}}=-\left(\frac{\Delta L}{L}+\frac{\Delta n}{n}\right)$$

Δ*L*, Δ*n*, Δ*f_light_*, and Δ*λ_light_* are the variations of cavity length, refractive index change caused by the photoelastic effect, the frequency shift, and the wavelength shift of the DBR fiber laser. For convenience, frequency variation is rewritten in the form of a relative wavelength shift as Equation (6).
(6)$$\frac{\Delta \lambda_{light}}{\lambda_{light}}=-\frac{\Delta f_{light}}{f_{light}}=\left\{1-\frac{n^{2}}{2}\left[p_{12}-\mu \left(p_{11}+p_{12}\right)p_{11}\right]\right\}\varepsilon$$

According to Hooke’s law, the strain tensor of the DBR fiber laser in a uniform sound field can be expressed as:(7)$$\varepsilon_{x}=\frac{(2\mu -1)P}{E}$$
where *E* is the Young’s modulus of the DBR fiber laser, and μ is the poisson ratio. By substituting Equation (7) into Equation (6), the relative sound pressure sensitivity *K*_DBR_ of the DBR fiber laser can be obtained:(8)$$K_{\mathrm{DBR}}=\frac{\Delta \lambda_{light}}{\lambda_{light}P}=\left\{1-\frac{n^{2}}{2}\left[p_{12}-\mu \left(p_{11}+p_{12}\right)p_{11}\right]\right\}\frac{(2\mu -1)P}{E}$$

As can be seen from Equation (8), the sound pressure sensitivity of the DBR fiber laser depends on the effective elastic coefficients *P*_11_ and *P*_12_. The Young’s modulus and Poisson ratio of the DBR fiber laser can be effectively improved by selecting appropriate materials for packaging.

### 2.2. Ultrasonic Wave Sensing Model of the DBR Fiber Laser

#### 2.2.1. The Optical Interrogation Signal of the DBR Fiber Laser

Assuming the operating wavelength of the DBR fiber laser is *λ*_0_ and the wavelength shift is Δ*λ*(*t*). The corresponding phase difference introduced by the NE-MZI has an expression as follows:(9)$$\Delta \phi \left(t\right)=\frac{2\pi nL_{OPD}}{\lambda_{0}^{2}}\Delta \lambda \left(t\right)$$
where *L_OPD_* is the optical path difference of the NE-MZI, and Δ*φ* (t) is the phase difference caused by the wavelength shift of the DBR fiber laser, i.e., the demodulation signal recorded in the experiments. The ultrasonic response comparison of the DBR fiber laser with the ultrasonic transducer takes advantage of the proportional relation between the applied sound pressure and the output voltage of the ultrasonic transducer under the same frequency and pressure conditions. The DBR fiber laser is placed right in front of the receiving transducer, facing the transmitting transducer. The phase difference proves to be proportional to the sound pressure applied to the DBR fiber laser. The demonstration scheme and results are plotted in Figure 4. 

Figure 4a is the experimental scheme of the demonstration experiment, and Figure 4b shows the influences of the ambient pressure on the ultrasonic noise power of the DBR fiber laser output. The noise power increases with the ambient pressure and is much smaller than the ultrasonic signal power. Noise will affect the accuracy of ultrasonic measurement, so the noise signal is subtracted during ultrasonic measurement. Figure 4c,d are the sample experimental spectra of phase difference at transducer output voltages for 25 kHz and 300 kHz ultrasonic waves. Figure 4 shows the phase difference of the DBR fiber laser changes linearly with the ultrasonic pressure, and the linear correlation coefficients of the experimental results are over 0.99.

#### 2.2.2. Calibration of the Ultrasonic Response of the DBR Fiber Laser

Since it is the first time to measure the ultrasonic sound pressure quantitatively using a DBR fiber laser, the ultrasonic response of the proposed DBR fiber laser system needs to be tested and compared with its corresponding transducers at the five ultrasonic frequencies of our applications. 

Figure 5 shows the optical responses of the DBR fiber laser to five sampling ultrasonic waves between 20 and 300 kHz at 25, 40, 112, 200, and 300 kHz, where the ultrasonic waves launched at five frequencies are tested simultaneously. The inlet plots the background noise without an ultrasonic signal applied.

The background noise power is relatively large at lower frequencies (for example, rising to −50 dB/Hz^1/2^ below 1 kHz), decreases with frequency increase, and drops below −75 dB/Hz^1/2^ after 15 kHz, where the noise equivalent sound pressure (NESP) reaches about −80 dB/Hz^1/2^. However, the noise power rises to −60 dB/Hz^1/2^ at 20 kHz as the ultrasonic wave launches. Therefore, calibration is necessary for the DBR fiber laser before sensing applications.

Five piezoelectric transducers of 25 kHz, 40 kHz, 112 kHz, 200 kHz, and 300 kHz are chosen for the sensitivity calibration of the DBR fiber laser. The results are shown in Figure 6.

Figure 6 shows the sensitivity of the DBR fiber laser and transducers over the frequency range from 25 to 300 kHz. The average sensitivity of the DBR fiber laser is −45.4 dB and relatively flat over the ultrasonic range. By now, the DBR fiber laser system is ready for ultrasonic detection and gas sensing.

## 3. Measurement of the Acoustic Relaxation Absorption and Concentration of CO_2_ with the DBR Fiber Laser Ultrasonic Sensor

### 3.1. Measurement of the Acoustic Relaxation Absorption at a Specific Frequency

Relaxation frequency and the corresponding intensity of the absorption curve are the unique characteristics of a specific gas and are generally used to identify the unknown composition of a gas mixture. The composition or concentration of a certain gas in the mixture can be identified with the sound velocity spectrum or the absorption spectrum, which depends on the number of components in the mixture. Characteristic frequencies, absorption coefficients, and sound velocity of the measured gas can be evaluated from the comparison of the reconstruction spectrum with the theoretical or published experimental ultrasonic spectral data of gases [18]. Since the transmitting transducers have no substitutes at present, the experiment uses a decompression gas chamber to simulate the main molecular relaxation process. 

Figure 7 illustrates the experimental system proposed in this paper, which includes two main units: a pressure-tunable gas chamber with two ultrasonic transducers of 25 and 40 kHz and a DBR laser sensor, and an interrogation system based on a NE-MZI and 3 × 3 coupler method. The pressure-tunable gas chamber, ranging from 0.01 to 1 atm, is designed for the measurement of the acoustic absorption spectrum. In the gas chamber, two transmitting transducers, embedded in a plate fixed on a step motor to adjust the distance between the transducers and the DBR fiber laser, emit ultrasonic waves of 25 and 40 kHz, which are received by the DBR fiber laser. The DBR fiber laser is placed on the path and axially perpendicular to the direction of the ultrasonic emission. Since the diameter of an optical fiber is much smaller than the ultrasonic wavelength, reflection from the fiber is not a concern in our experiments. An aluminum plate is placed next to the DBR fiber laser on the opposite side of the transmitting transducer to form an effective standing wave. 

The reflected output signal of 1550 nm of the DBR fiber laser, which is pumped by a 980 nm laser diode through port 1 of a wavelength division multiplexer (WDM), passes the WDM again from port 2–3 and a 2 × 2 coupler, feeds into a NE-MZI followed by a 3 × 3 coupler, and is finally received and converted into electrical signals by photodetectors (PDs) and processed with a data acquisition card (DAQ). A very small phase difference and corresponding wavelength shift of the DBR fiber laser under an applied ultrasonic wave can be discriminated by the NE-MZI with a 100 m optical path difference and the 3 × 3 coupler method, which effectively improves the sensitivity of the system.

CO_2_ is a popular validation gas in reduction studies of the acoustic relaxation absorption spectrum. Acoustic absorption measures are first used for the reconstruction of the acoustic relaxation absorption curve of CO_2_. According to the Clapeyron equation, a decreasing chamber pressure is equivalent to an increasing ultrasonic frequency. Since the effective relaxation frequency of the acoustic relaxation absorption curve of 100% CO_2_ is located at 40 kHz, an ultrasonic transducer of 25 kHz is a better choice to satisfy the measurement of the relaxation absorption spectrum of CO_2_, where f/p is controlled by adjusting the chamber pressure. The distance between the transmitting transducer and the DBR fiber laser varies between 30~100 mm to satisfy the far-field prerequisite for ultrasonic attenuation coefficient measurement. 

The incidence ultrasonic wave is reflected on the receiving transducer/aluminum plate and interferes with its reflected counterpart; thus, the sound field is a superposition of sound absorption and interference. According to the classical definition of sound absorption coefficient in Equation (10), acoustic pressure needs to be obtained along the direction of ultrasonic emission at various distances, where the transmitting transducer moves on a linear guide rail controlled by a step-motor from 30 mm (distance to the DBR fiber laser) to 100 mm and records the output phase difference of the DBR fiber laser sensor every 1 mm.
(10)$$\alpha =\frac{\ln U_{1}-\ln U_{2}}{L_{2}-L_{1}}=\frac{\ln k_{t}p_{1}-\ln k_{t}p_{2}}{L_{2}-L_{1}}=\frac{\ln p_{1}-\ln p_{2}}{L_{2}-L_{1}}$$
where *α* is the absorption coefficient and *U*_1_ and *U*_2_ are the output voltages of the receiving transducer obtained in positions *L*_1_ and *L*_2_, respectively. The output voltage *U* is also defined as a product of sound pressure sensitivity *k_t_* and intensity *P*. Equation (10) clarifies that the absorption coefficient *α* is independent of the sensitivity of the sensor *k_t_* for a specific ultrasonic frequency and only determined by the gas characteristics. Therefore, the absorption coefficient *α* stays the same no matter whether a transducer or a DBR fiber laser sensor is used for testing, as long as the f/p stays the same. The sensitivity determines the lowest detectable acoustic pressure, or the noise equivalent sound pressure, so the measurement precision of the acoustic attenuation coefficient improves with the sensitivity.

Testing results of the 25 kHz sound field distribution recovered by the DBR fiber laser sensor are plotted in Figure 8a. Based on the linear relation of phase difference and acoustic pressure and Equation (10), a diagram of peak pressure level in Figure 8a to the distance and its linear fitting are shown in Figure 8b, and the slope of which is divided by 20 is equal to the absorption coefficient of the gas, where the phase difference variation limits to a linear zone because of the weak ultrasonic absorption.

From Figure 8a, the standing sound wave field produced by the superposition of the incident and reflected ultrasonic waves shows a tendency toward oscillation decline. According to Equation (10) and the direct conversion relationship between sound pressure and sound pressure level, the slope of the fitting line in Figure 8b divided by 20 is the acoustic absorption coefficient α and equals 0.01 dB/mm. The ultrasonic wavelength *λ* is 11.3 mm, and the acoustic relaxation absorption coefficient *αλ* is 0.113. Following the same process to get the acoustic relaxation absorption coefficients at various frequencies by tuning the gas chamber pressure, the acoustic relaxation absorption spectrum is recovered. 

### 3.2. Measurement Results of the Absorption Spectrum of CO_2_

To restore the gas relaxation absorption spectrum, ultrasonic attenuation coefficients over a wide range of f/p need to be obtained through a series of measurements with various pressures. The acoustic attenuation increases as the pressure in the gas chamber decreases. The minimum detectable sound pressure of the DBR fiber laser sensor determines the minimum chamber pressure in the experiments. A smaller detectable sound pressure means even lower applicable chamber pressure and a wider frequency bandwidth, and the reconstruction of the acoustic relaxation absorption spectra gets more precise.

A group of experiments are carried out with the gas chamber pressure tuned between 0.1 and 1.0 atm for CO_2_ concentrations of 100%, 80%, 50%, and 20%. Measurements use two ultrasonic transducers of 25 kHz and 40 kHz as the emitting sources and the DBR fiber laser as the sensing probe. Repeating the test steps in Section 3.1, acoustic relaxation absorption coefficients are obtained corresponding to a series of f/p around the effective relaxation frequency. The acoustic relaxation absorption spectrum of 100% CO_2_ is restored by the fitting spectrum of 25 kHz (red square) and 40 kHz (pink round), as shown in Figure 9a. Similar fitting curves are plotted in Figure 9b–d for CO_2_ concentrations of 80%, 50%, and 20%, respectively. The black solid curves are theoretical curves corresponding to the four CO_2_ concentrations.

In Figure 9a, the experimental acoustic relaxation absorption spectrum slightly deviates from the theoretical spectrum of 100% CO_2_, which leads to an error of 0.82% of concentration. In Figure 9b, the deviation of the experimental spectrum from the theoretical spectrum leads to an error of 0.84%. For 50% and 20% CO_2_, the errors are 1.26% and 1.32%, respectively. While the relaxation absorption coefficient decreases with the concentration of CO_2_, the measurement error of the corresponding acoustic relaxation absorption spectra increases. On the other side, it shows that measurements taken as close as possible to the absorption peak frequency will give better precision.

## 4. Discussion

### 4.1. Systematic Error Results from Variation of f/p

The deviation of the experimental results from the theoretical model increasing with f/p, i.e., higher frequency, might stem from several factors. First, the acoustic impedance mismatch of the transducer in gas affects both acoustic generation and detection and results in a decreasing signal-to-noise ratio (SNR) with increasing frequency. Second, any foreign molecules present in the gas chamber (for example, gas leaking from the vacuum pump) may induce multiple molecular relaxation pathways, leading to a possible broadening of the acoustic relaxation absorption spectrum. These issues might also be responsible for the generally larger standard deviations at low pressures. In general, raising the f/p ratio leads to a larger measurement error of the relaxation absorption coefficient.

### 4.2. The Temperature and Pressure Instability of the Gas Chamber and Compensation

Pressure instability in the gas chamber is mainly related to poor tightness and circumstantial temperature variation. The air tightness of the gas chamber affects not only the f/p ratio during measurement but also the acoustic relaxation absorption coefficient. When the gas chamber pressure drops below 1 kPa, the average pressure variation in the gas chamber is about 3 Pa/10 min, where 10 min is the time of one complete test, indicating an error of the f/p ratio caused by the gas chamber pressure increasing by about 0.003% every 10 min. During the measurement of the acoustic relaxation absorption coefficient, the temperature varies less than 0.01 K every 10 min in the gas chamber, resulting in a sound velocity error of about 0.002% and a f/p error of 0.033%. Therefore, the temperature was compensated for at 293 K, and the pressure of the gas chamber was corrected accordingly.

## 5. Conclusions

In brief, a novel optical approach using a DBR fiber laser is proposed to measure the acoustic absorption coefficient over a wide frequency band and apply reconstruction of the acoustic relaxation absorption spectra of CO_2_ with various concentrations. With a relative wide and flat frequency response, the DBR fiber laser sensor measures and restores a full acoustic relaxation absorption spectrum of CO_2_ using a decompression gas chamber between 0.1 and 1 atm to accommodate the main molecular relaxation processes and interrogates with a NE-MZI to gain a sound pressure sensitivity of −45.4 dB. The measurement error of the acoustic relaxation absorption spectrum is less than 1.32%. Combined with the gas sensing approach based on ultrasonic velocity, the DBR fiber laser sensor is a promising candidate for the analysis of ternary or quaternary gases.

## Figures and Tables

**Figure 1 sensors-23-04740-f001:**
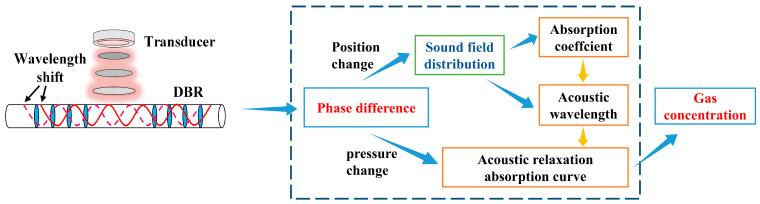
Flow diagram of ultrasonic gas sensing using the DBR fiber laser.

**Figure 2 sensors-23-04740-f002:**
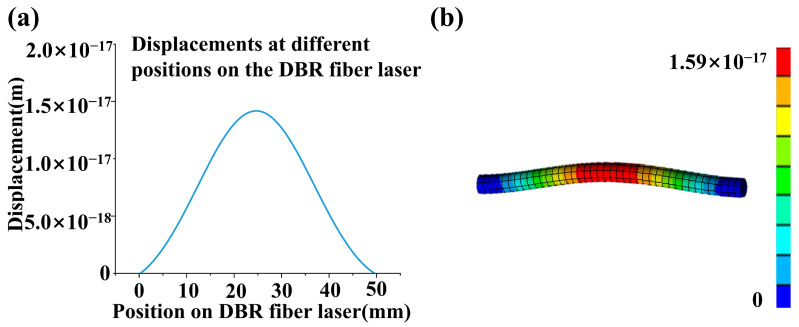
Displacement distribution along the DBR fiber laser under 25 kHz and a 2 Pa ultrasonic wave: (**a**) solution of the vibration equation and (**b**) finite element simulation.

**Figure 3 sensors-23-04740-f003:**
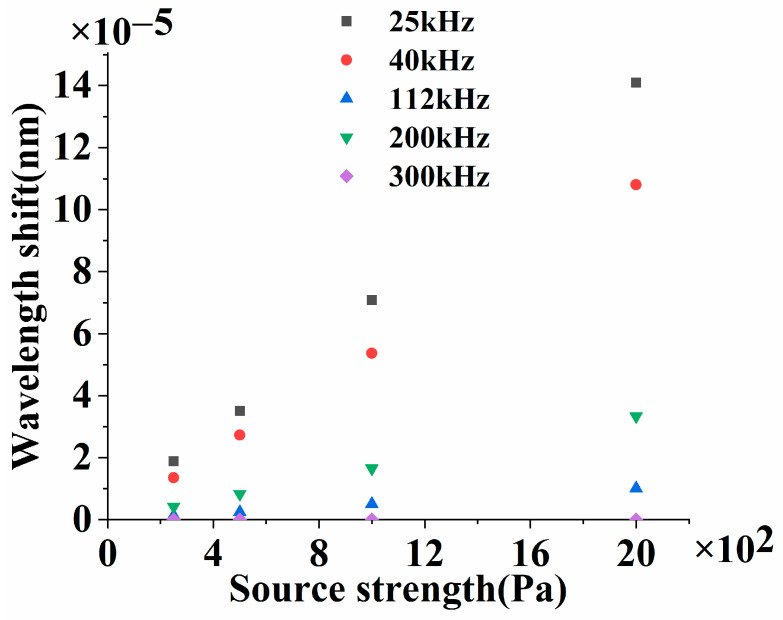
Corresponding wavelength shift of the DBR fiber laser to the source strength of ultrasonic waves at various frequencies.

**Figure 4 sensors-23-04740-f004:**
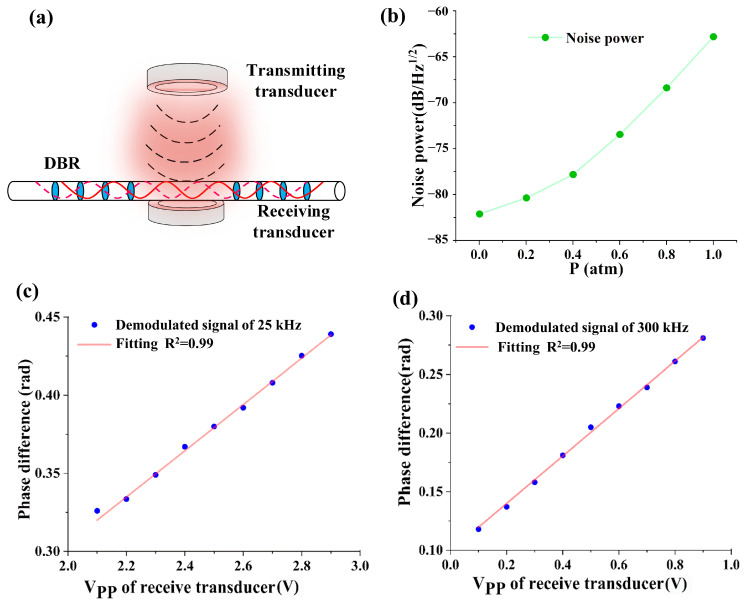
(**a**) Schematic diagram of linearity demonstration with a DBR fiber laser and a pair of transducers; (**b**) experimental results of the noise power to ambient pressure in the gas chamber; (**c**,**d**) are sample ultrasonic calibration results at 25 kHz and 300 kHz using the DBR fiber laser and a receiving transducer.

**Figure 5 sensors-23-04740-f005:**
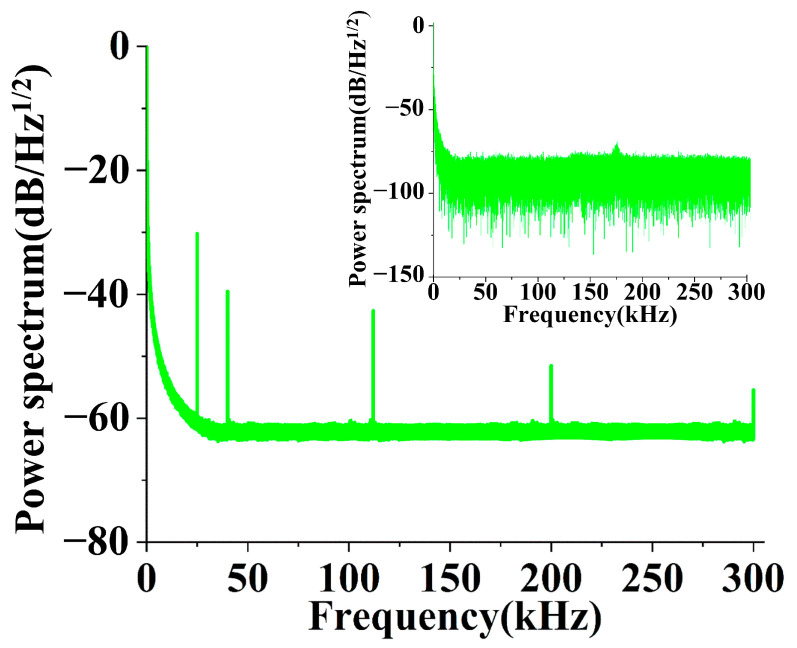
Optical background phase noise spectrum and power spectrum of the DBR fiber laser.

**Figure 6 sensors-23-04740-f006:**
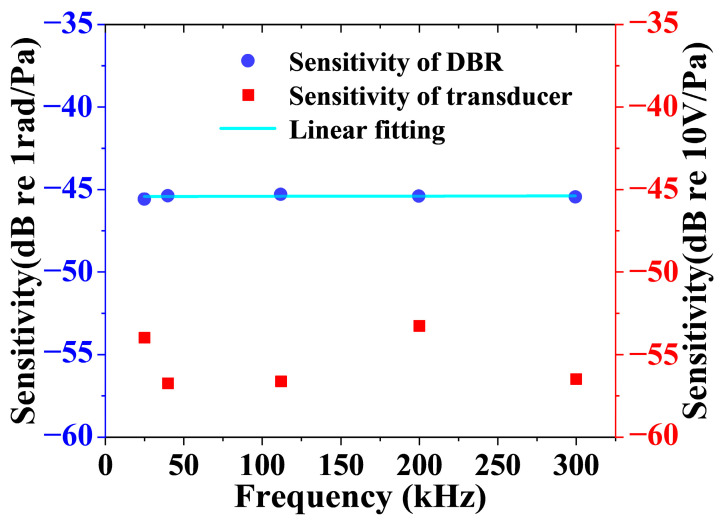
Sensitivity testing results of a DBR fiber laser at five different frequencies.

**Figure 7 sensors-23-04740-f007:**
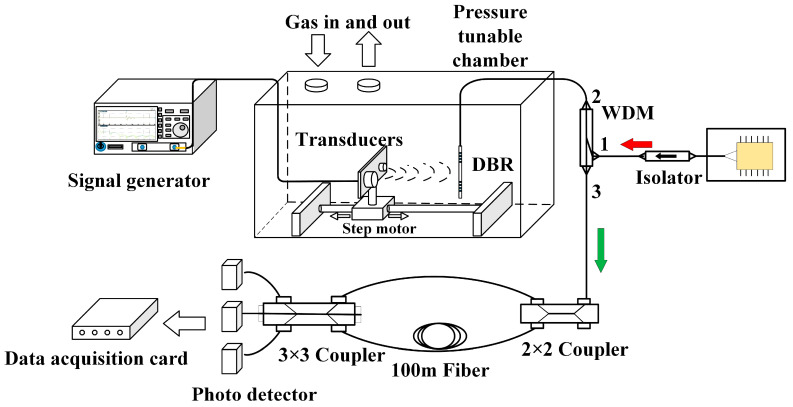
Diagram of the DBR fiber laser gas detection system.

**Figure 8 sensors-23-04740-f008:**
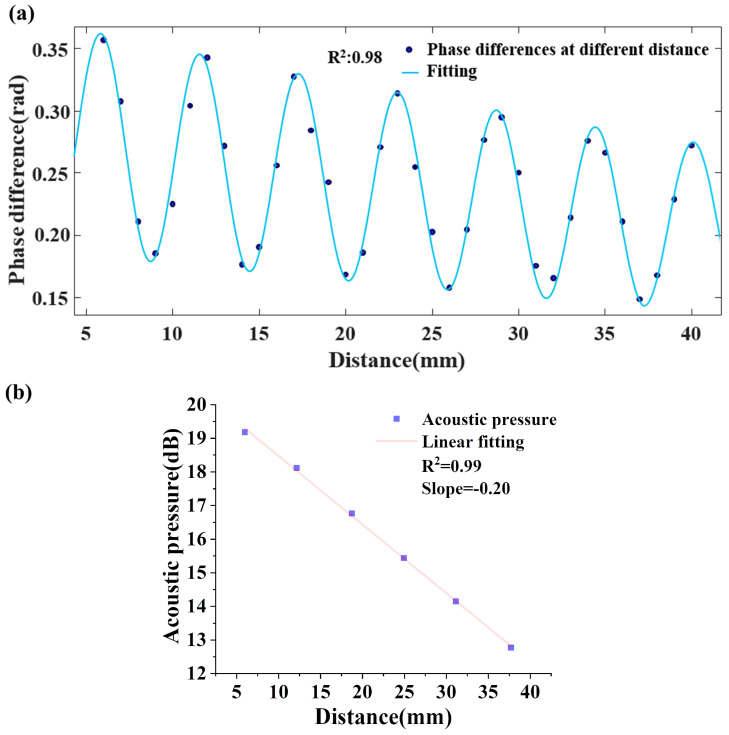
The testing results of the DBR fiber laser sensor. (**a**) The ultrasonic sound field distribution of 25 kHz in 100% CO_2_ and (**b**) the acoustic pressure to distance curve.

**Figure 9 sensors-23-04740-f009:**
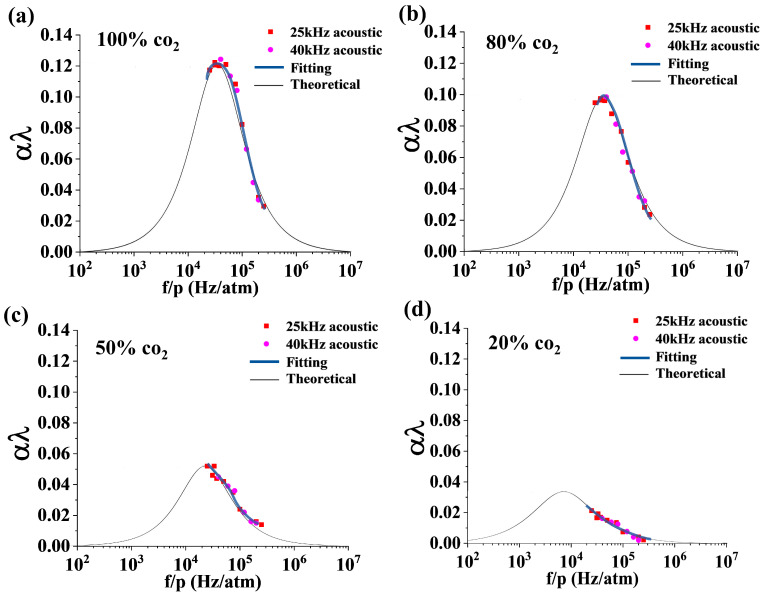
Restored curves of the acoustic absorption relaxation (pink fitting curve) with the DBR fiber laser and the theoretical curve (black solid curve) [18] of CO_2_, concentrations of (**a**) 100%, (**b**) 80%, (**c**) 50%, and (**d**) 20%. Using both 25 kHz and 40 kHz ultrasonic transducers, testing in the gas chamber over a pressure range of 0.1 to 1 atm.

## Data Availability

The data presented in this study are available on request from the corresponding authors.

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
