# Peer review of "Measurement of the Acoustic Relaxation Absorption Spectrum of CO_2_ Using a Distributed Bragg Reflector Fiber Laser"

_sensors, 2023, doi:10.3390/s23104740_

Round 1
Reviewer 1 Report
The researchers demonstrated a method that can measure the gas concentrations based on acoustic relaxation absorption spectrum and optical fiber distributed Bragg reflector laser. The topic is interesting and the discussions is solid. Some suggestions have been raised to help further improve the quality of the paper.
1. Please use the full name of DBR in the title.
2. In line 42, what is the difference between classical absorption and the acoustic relaxation absorption?
3. In Fig.3, is the unit of source strength correct? Is this unit an international standard unit?
4. In Line 221, the font size of noise power is different from others.
5. In Line 250,the researchers used 25 kHz to 300 kHz, is it possible to use MHz frequency?
6. In Fig.7, please provide the full name of PD and DAQ in either the figure caption or the main text.
7. In section 3, is it possible to demonstrate this method using several mixed gases?
8. In daily life, it is hard to find a environment where there is only gas existing. In most environment gas, liquid, and solid exist together. In this case, does this system and method work well? Please explain the reasons.
Reviewer 2 Report
In this paper, the authors present a wideband ultrasonic sensor using a distributed Bragg reflector (DBR) fiber laser for gas concentration detection through acoustic relaxation absorption curve reconstruction. With a relatively wide and flat frequency response, the DBR fiber laser sensor measures and restores a full acoustic relaxation absorption spectrum of CO2 using a decompression gas chamber between 0.1 and 1 atm to accommodate the main molecular relaxation processes, and interrogates with a non-equilibrium Mach-Zehnder (M-20 Z) interferometer to gain a sound pressure sensitivity of -45.4 dB. This article is clear, concise, and suitable for the scope of the journal. Several small suggestions are supplied:
1. Suggest the authors supply more detail about the experiment setup.
2. Suggest the authors separate the discussion and conclusion parts.
3. Marine structure monitoring also promising based on optical fiber sensing technology, suggest the authors enhance the introduction part with one last review on this topic:
Optical fiber sensing for marine environment and marine structural health monitoring: A review Optics and Laser Technology, 2021.
Minor editing of English language required.
